# ZBED1 Regulates Genes Important for Multiple Biological Processes of the Placenta

**DOI:** 10.3390/genes13010133

**Published:** 2022-01-12

**Authors:** Simone Johansen, Sofie Traynor, Malene Laage Ebstrup, Mikkel Green Terp, Christina Bøg Pedersen, Henrik Jørn Ditzel, Morten Frier Gjerstorff

**Affiliations:** 1Department of Cancer and Inflammation Research, Institute for Molecular Medicine, University of Southern Denmark, 5000 Odense, Denmark; sijohansen@health.sdu.dk (S.J.); sofiet@health.sdu.dk (S.T.); male@cancer.dk (M.L.E.); mterp@health.sdu.dk (M.G.T.); cbpedersen@health.sdu.dk (C.B.P.); hditzel@health.sdu.dk (H.J.D.); 2Department of Oncology, Odense University Hospital, 5230 Odense, Denmark; 3Academy of Geriatric Cancer Research (AgeCare), Odense University Hospital, 5230 Odense, Denmark

**Keywords:** ZBED1, trophoblast differentiation, BeWo

## Abstract

The transcription factor ZBED1 is highly expressed in trophoblast cells, but its functions in the processes of trophoblast and placental biology remain elusive. Here, we characterized the role of ZBED1 in trophoblast cell differentiation using an in vitro BeWo cell model. We demonstrate that *ZBED1* is enhanced in its expression early after forskolin-induced differentiation of BeWo cells and regulates many of the genes that are differentially expressed as an effect of forskolin treatment. Specifically, genes encoding markers for the differentiation of cytotrophoblast into syncytiotrophoblast and factors essential for trophoblast cell fusion and invasion were negatively regulated by ZBED1, indicating that ZBED1 might be important for maintaining a steady pool of cytotrophoblast cells. In addition, ZBED1 affected genes involved in the regulation of trophoblast cell survival and apoptosis, in agreement with the observed increase in apoptosis upon knockdown of *ZBED1* in forskolin-treated BeWo cells. In addition, genes implicated in the differentiation, recruitment, and function of innate immune cells by the placenta were affected by ZBED1, further suggesting a role for this protein in the regulation of maternal immune tolerance. In conclusion, our study implicates ZBED1 in major biological processes of placental biology.

## 1. Introduction

The human zinc finger BED domain-containing protein 1 (ZBED1), also known as the human DNA replication-related element-binding factor (hDREF), is a transcription factor suggested to regulate cell cycle progression and proliferation [1,2]. ZBED1 consists of two conserved domains; a C-terminal ATC domain important for self-association and nuclear accumulation and an N-terminal BED zinc finger domain with DNA binding activity directed to the consensus sequence 5′-TGTCG(C/T)GA(C/T)A [1,3]. Initial database searches have mapped the binding sequence of ZBED1 to various promoters of genes involved in DNA replication and repair, cell cycle regulation, and transcription [1], suggesting that ZBED1 is required to maintain the basal transcription machinery and is important for normal cell growth and proliferation. Accordingly, *ZBED1* expression is induced in the G1-S phase and regulates the expression of genes such as histone 1 (H1) and ribosomal proteins [1,4], which might be facilitated by regulating the nucleosome remodeling and deacetylase complex (NuRD) [5]. In addition, a recent study has elucidated a link between *ZBED1* overexpression in gastric cancer and poor prognosis and further demonstrated that ectopic expression of ZBED1 promotes cell proliferation and colony formation abilities of gastric cancer cell lines [6]. We have demonstrated that ZBED1 is also expressed in non-dividing cells, providing evidence that ZBED1 might be involved in biological processes other than cell proliferation [7]. ZBED1 expression diminished with the maturation of epithelial cells and was abundantly expressed in the placenta [7], suggesting that ZBED1 might play a role in cellular differentiation and fetal development. Similarly, the *Drosophila* homologue of ZBED1 (dDREF) has been implicated in different aspects of *Drosophila* development [8,9,10,11,12]. Hence, investigation of ZBED1 in developmental contexts of the human organism, such as the placenta, might elucidate additional functions of ZBED1.

In the placenta, cytotrophoblasts fuse and differentiate into syncytiotrophoblasts, which form the syncytium or differentiate into extravillous trophoblasts [13,14,15]. Syncytiotrophoblast differentiation has been addressed in several in vitro studies focusing on two different aspects, i.e., morphological differentiation and biochemical differentiation. The former involves the fusion of cytotrophoblasts to form a syncytium, and the latter involves the expression of genes involved in cell-cell fusion, fetal-maternal exchange, and immunoregulation, hormone synthesis, and secretion, as well as other functions of terminally differentiated syncytiotrophoblasts [13]. Essentially this includes expression of fusogenic genes, such as syncytin-1 and syncytin-2 [16,17,18], genes likely to play a role in membrane repairs such as dysferlin [19], genes involved in hormone synthesis such as *CYP19* [20], and genes encoding hormones such as estrogens and human chorionic gonadotropin (*hCG*) [13,21,22]. These genes are all regulated downstream of the earliest-acting transcription factor known to induce syncytialization, termed glial cell missing 1 (*GCM1*) [22,23].

Maternal immune tolerance is crucial to protect the fetus and fetal-derived tissue from immunological recognition and rejection by the maternal immune system [24]. The presence of immune cells is essential to modulate an environment that favors decidua and spiral artery remodeling and trophoblast invasion, important for a successful pregnancy [25]. To do so, decidual stromal cells and trophoblasts secrete cytokines and generate chemokine gradients that actively recruit maternal leukocytes to the fetal-maternal interface [26] and subsequently modulate their function by polarizing macrophages to an M2-like phenotype, inhibiting uterine natural killer cell cytotoxicity, inducing tolerogenic dendritic cells and differentiating regulatory T cells, among others [27,28]. This also includes the release of apoptosis mediators and inhibitors from syncytiotrophoblast-derived vesicles [29] that dampen the cytotoxic effect of immune cells [30,31].

To further understand the biological functions of ZBED1, we characterized the role of this protein during trophoblast differentiation and fusion. We used the BeWo cell line model, which has a naturally high expression of ZBED1 [32] and is able to differentiate and fuse upon chemical stimuli with forskolin [33,34]. Our results implicate ZBED1 in important processes of placental biology.

## 2. Materials and Methods

### 2.1. BeWo Cell Culture

The human choriocarcinoma cell line BeWo (Sigma-Aldrich, Søborg, Denmark) was purchased from ATCC and cultured in HAM’s F12 medium (Thermo Fisher Scientific, Roskilde, Denmark) supplemented with 10% fetal bovine serum (Sigma-Aldrich), 1% penicillin/streptomycin (Sigma-Aldrich), and 1× Glutamax (Thermo Fisher Scientific). Cells were kept at low passage and regularly tested for mycoplasma (MycoAlert, Mycoplasma detection kit, Lonza, Vallensbæk Strand, Denmark). To induce differentiation and fusion, the cells were treated with media containing 20 μM of forskolin (Sigma-Aldrich). Control cells were treated with vehicle (DMSO). Media was renewed every 24–48 h. At the onset of experiments, the cells were seeded in densities of 51,790 cells/cm^2^.

### 2.2. ZBED1 shRNA Knockdown

ZBED1 and luciferase (negative control) shRNA constructs (shZBED1 and shLuc) were prepared as lentivirus using pSicoR PGK puro vectors (Addgene plasmid ID 12084). The shRNA sequence for ZBED1 (5′-GTGGCCATGTACATGCTCTAT) and luciferase (5′-CGCTGAGTACTTCGAAATGTC) were cloned into the cloning site of the vector, and the vectors were prepared as lentivirus by cotransfection with pMD2.G, pRSV-Rev, and pMDL g/p RRE (kindly provided by the Trono Lab through Addgene, Cambridge, UK) into HEK293T cells using PEI. After 20–24 h, the medium was changed to fresh DMEM medium with additives (10% FBS and 1% penicillin/streptomycin), and 48 h thereafter, the virus was harvested, filtered, and precipitated with PEG before resuspension in HAM’s F12 medium. For knockdown experiments, the virus was titrated to determine the amount needed for optimal *ZBED1* knockdown and similar viral expression levels of shZBED1 and control shRNA by measuring the relative expression of *ZBED1* and 5′-LTR 72 h after virus infection. For knockdown experiments, cells were seeded and the next day transduced with the respective shRNA construct in medium with the addition of 6 μg/mL of polybrene (Sigma-Aldrich). After 24 h incubation, the medium was changed to fresh medium, and after another 24 h, the transduced cells were set up for experiments.

### 2.3. Western Blotting

Cells were harvested with trypsin and proteins extracted in RIPA buffer with protease inhibitors (Complete Protease Inhibitor Cocktail, Sigma-Aldrich). The extracts were treated with benzonase nuclease (Sigma-Aldrich), and protein concentration was determined using the Pierce BCA Protein Assay Kit (Thermo Fischer). Lysates were subjected to gel electrophoresis on a 12-well 4%–20% SDS-page gel (Biorad, cat. no.: 4561095; Copenhagen, Denmark) and electroblotted onto a PVDF membrane. For blocking of remaining binding sites, the membranes were incubated in PBS with 0.1% Tween-20 and 5% non-fat dry milk powder for Western blotting with anti-ZBED1 (1:2000, Sigma, wh0009189M1) or TBS with 0.1% Tween-20 and 5% non-fat dry milk powder for Western blotting with anti-α/β tubulin (1:10,000, Cell signaling, 2148S) for control. The membranes were further stained with horseradish peroxidase-conjugated goat anti-mouse (for anti-ZBED1, 1:5000) or goat anti-rabbit (for anti-α/β tubulin, 1:5000) (DakoCytomation Denmark A/S, Glostrup, Denmark) and developed with ECL substrate solution A and B (Biorad, Copenhagen, Denmark).

### 2.4. Immunocytochemical Staining and Fluorescence Microscopy

BeWo cells were treated with forskolin or DMSO for 0, 24, and 72 h before fixation with ice-cold 100% methanol for 20 min at −20 °C and permeabilization in 0.2% Triton X-100 (Sigma) in PBS for 10 min. After blocking with 3% BSA in PBS for 30 min, the coverslips were incubated with mouse anti-zonulae occludens-1 (ZO-1; BD Transduction Laboratories, Z72720), diluted 1:200 in 3% BSA (Sigma, cat. no.: A7906) in PBS for 90 min at room temperature. ZO-1 is a tight junction protein visualizing cell-cell boundary. The cells were then washed in PBS and incubated with goat anti-mouse Alexa Fluor 488 diluted 1:500 in 3% BSA (Sigma, cat. no.: A7906) in PBS for 1 h. For the last 5 min of incubation, 1 μL/mL DAPI (Sigma-Aldrich, cat. no.: D9542) was added for staining of the cell nucleus. Finally, the cells were washed in PBS before mounting on glass slides using ProLong Gold Antifade Mountant without DAPI (Fischer Scientific, cat. no.: P36930). Images were obtained using an Olympus IX73 microscope.

### 2.5. Growth Assay

Apoptosis of shLuc- and shZBED1-transduced BeWo cells treated with forskolin or DMSO for 0, 24, and 72 h were measured using the Cell Death Detection Elisa Plus kit (Roche, cat. no.: 11774425001) according to the manufacturer’s protocol. The results were normalized to the number of cells at the respective time points.

### 2.6. DNA Extraction for Quantifying Fusion

At 0, 24, and 74 h of forskolin or DMSO treatment, BeWo cells were harvested in trypsin and counted. The cells were pelleted by centrifugation for 5 min at 300× *g*, and DNA was purified using the DNeasy and Tissue Kit (Qiagen, cat. no.: 69504) according to the manufacturer’s protocol. DNA concentration, A260/A280, and A260/A230 absorbances were measured with a NanoDrop spectrophotometer (Thermo Scientific NanoDrop One), and the average DNA/cell content was calculated.

### 2.7. Quantitative PCR

RNA was purified using TriZol (Thermo Fischer) chloroform RNA extraction. A total of 1 µg of DNAse-treated RNA was used for cDNA synthesis with Maxima H Reverse Transcriptase (Thermo Fischer). Relative quantification of gene expression was performed with 5 µL of cDNA mixed with SYBR green polymerase chain reaction mastermix (Qiagen). Primers specific for ZBED1 (Qiagen, QT00224140), PUM1 (Qiagen, QT00029421), 5′-LTR (5′-TGGCTAACTAGGGAACCCACTGCT, 3′-TCACACAACAGACGG GCACACAC), GCM1 (5′-TGGGACTTGAACCAGCAGTAAG, 3′-CAGGAGATTGTT TTCTAGGGCTTCT), hCG-β subunit 3 (hCGβ) (5′-CTACTGCCCCACCATGACCC, 3′-GATGGACTCGAAGCGCACAT), dysferlin (5′-TATGCCGA GAACGTCCACAC, 3′-TCTTCACCCCTGCAAACACC), and syncytin-1 (5′-TCCGTACCCATACTCGCCTG, 3′-AGTAGGGTTTTGGGCCGAGA) were used. The qPCR data were analyzed using the comparative C_T_ method in the StepOne software (version). Mean CT values were normalized to PUM1 and (2^-(CT_primer/CT_PUM1)^).

### 2.8. Statistical Testing

The fold change of the samples is represented relative to the control sample at 0 h. The experiments having three biological replicates were analyzed by two-way analysis of variance (ANOVA) as both treatment/transduction and time point were considered as factors. Multiple comparison test was used to determine which groups differed from each other by comparing the mean of the treatments/transduction to every other mean at the same time point based on a 5% significance level. Adjusted *p*-values are shown in this study.

### 2.9. RNA Sequencing

The mRNAs of three biological replicates of BeWo cells, either transduced with shLuc or shZBED1, upon 0, 24, and 72 h of forskolin treatment, were subjected to RNA sequencing. RNA purity and the total amount of RNA were quantified on a NanoDrop 2000 spectrophotometer. Briefly, the samples were prepared using the TrueSeq RNA sample preparation kit (Illumina, Copenhagen, Denmark) and sequenced on a NovaSeq 6000. The RNA-seq reads were aligned to the human genome (hg19) using the Spliced Transcripts Alignment to a Reference (STAR) software [35] with default parameters, and tags in exons were counted using iRNA-seq [36]. The final exon count matrix was analyzed for differential gene expression using the DESeq2 pipeline in RStudio version 3.6.3 [37]. The exon count matrix was normalized by log transformation with respect to library size, and the reads were calculated as reads per kilobase to calculate the relative number of mRNA copies per gene. Differentially expressed genes were defined as genes with FDR-adjusted *p*-values ≤ 0.05 and log2 fold change above and below 1 and −1, respectively. Functional enrichment analysis of the differentially regulated genes was performed using the *HOMER* gene ontology analysis platform [38] to elucidate biological processes of the trophoblast differentiation and alterations of this model by *ZBED1* knockdown.

## 3. Results

### 3.1. ZBED1 Expression Is Upregulated during Trophoblast Cell Differentiation

To investigate the role of ZBED1 in trophoblast cell differentiation, we initially established a forskolin-induced BeWo cell fusion model. As expected, treatment of BeWo cells with 20 µM forskolin for 24 or 72 h caused significant upregulation of the relative expression of GCM1, hCGβ, syncytin-1, and dysferlin in a time-dependent manner, reaching fold upregulation of 4.5 (GCM1; Figure 1A), 33.5 (hCGβ; Figure 1B), 3.6 (syncytin-1; Figure 1C), and 11.9 (dyserflin; Figure 1D) when compared to DMSO control at 72 h. Additionally, when BeWo cells were treated with forskolin, cells exhibited larger nuclei that clustered in multi-nucleate syncytia with loss of the tight junction protein ZO-1 (Figure 1E). In contrast, untreated BeWo cells showed a distinct pattern of uninuclear cells with prominent ZO-1 staining at the epithelial borders, and only rarely were fused cells observed. Together, these observations demonstrate that forskolin is a potent inducer of BeWo differentiation and fusion. Interestingly, forskolin treatment of BeWo cells upregulated the expression of ZBED1 (24 h; 3.2-fold, 72 h; 5.5-fold, Figure 1F), suggesting that ZBED1 might have a regulatory role during BeWo differentiation.

### 3.2. ZBED1 Regulates the Expression of Trophoblast Differentiation Markers

To further evaluate the role of ZBED1 in trophoblast differentiation, we established a *ZBED1* knockdown model in which BeWo cells were transduced with shRNA targeting *ZBED1* or luciferase as control, and knockdown was validated by qPCR and Western blotting (Figure 2A,B, Appendix A). We evaluated the relative gene expression of syncytin-1, dysferlin, GCM1, and hCGβ before forskolin treatment (0 h) and after 24 and 72 h forskolin treatment of shLuc- (control) and shZBED1- (*ZBED1* knockdown) transduced BeWo cells. Interestingly, *ZBED1* knockdown increased the relative expression of syncytin-1 at 24 h (fold change 1,68; Figure 2C) and dysferlin after 24 and 72 h forskolin treatment in a time-dependent manner (fold change 1,8 and 2, respectively; Figure 2D) compared to the respective control cells. In contrast, *ZBED1* knockdown did not change the relative expression of GCM1 (Figure 2E) or hCGβ (Figure 2F). This suggests that ZBED1 may have an important role in regulating the expression of core trophoblast differentiation genes and that this was more prominent at 24 vs. 72 h of forskolin treatment. To further characterize the role of ZBED1 in trophoblast differentiation, the fusion of the cells with and without *ZBED1* knockdown was measured by calculating the DNA/cell ratio, as fused multinuclear cells comprise more DNA per cell than uninuclear cells. Upon forskolin treatment, both the control- and *ZBED1*-knockdown cells fused at 72 h (Figure 2G). Upon *ZBED1* knockdown, an increase in the relative gene expression of fusion-related genes (syncytin-1 and dysferlin) was observed, which was accompanied by a tendency toward an increase in the DNA/cell ratio compared to control cells at 72 h, although it did not reach significance (Figure 2G).

### 3.3. ZBED1 Is Important for the Viability of Trophoblast Cells during Differentiation

ZBED1 is known to be pro-proliferative [1,2,4], but the knockdown of *ZBED1* in BeWo cells did not affect cell growth (Figure 3A). As expected, BeWo cells stopped proliferating once syncytialization occurred upon forskolin stimuli, but upon concomitant knockdown of *ZBED1*, there was a reduction in cell numbers (Figure 3B). Furthermore, knockdown of *ZBED1* in BeWo cells under forskolin stimuli resulted in high levels of apoptotic cells, which was not observed in the absence of forskolin (Figure 3C,D). These results suggest that ZBED1 does not support the proliferation of BeWo cells but is required for BeWo cell viability during trophoblast differentiation.

### 3.4. ZBED1 Regulates Genes Important for Multiple Processes of Placental Biology

To further characterize the role of ZBED1 in BeWo cell differentiation, we performed RNA sequencing of BeWo cells with or without *ZBED1* knockdown upon forskolin treatment. Principal component analysis revealed a suitable correlation between the biological replicates, and the samples were clustered in five distinct groups based on *ZBED1* knockdown and forskolin treatment (Figure 4A). The only exception was cells treated with forskolin for 72 h hours, where cells with *ZBED1* knockdown and control cells were clustered together. This correlates well with the assumption that ZBED1 may be important early in trophoblast differentiation. Based on the number of regulated genes, *ZBED1* knockdown had only a modest effect on the gene expression of BeWo cells in the absence of forskolin (Figure 4B). However, the effect of *ZBED1* knockdown on gene expression was more significant upon forskolin-induced differentiation (Figure 4B). Interestingly, half of the genes upregulated by *ZBED1* knockdown (48 out of 81 genes) were also enhanced in their expression by forskolin, and 75 of the 168 genes in which expression was reduced by *ZBED1* knockdown were additionally reduced by forskolin (Figure 5A). The overlap between ZBED1- and forskolin-regulated genes was less significant at other time points (Appendix A). These results suggest that ZBED1 might counteract the early effects of forskolin-induced differentiation on the gene expression profile of BeWo cells.

Functional enrichment analysis of differentially regulated genes at 24 h showed that *ZBED1* knockdown affected pathways related to trophoblast differentiation, as well as proliferation and survival of cells (Figure 5B). Analysis of individual genes regulated by ZBED1 during the differentiation process further implicated ZBED1 in trophoblast and placental biology (Figure 5C). For instance, ZBED1 regulated genes essential for trophoblast fusion and estrogen synthesis (e.g., *CYP19A1, FER1L5,* and *ERVFDR-1*) and genes supporting trophoblast differentiation and invasion (e.g., *APELA*, *EID-2*, *ELF-5*, *ETV7*, *KLF8*, *NR6A1*, *BGN*, *HSPB1*, *HSPB8*, *INHA*, *KLF4*, *SGK1*, *TBX3*, and *TRERF1*).

Results from this and previous studies have implicated ZBED1 in the regulation of cell proliferation and apoptosis. In agreement, *ZBED1* knockdown induced the expression of genes regulating cell proliferation and apoptosis upon forskolin treatment (e.g., *GADD45G, RELT, DDIT3, EGR1, KL4, TNFRSF1B, PDGFRB, and ID-1)* (Figure 5C). Finally, *ZBED1* knockdown downregulated genes encoding factors implicated in the maintenance of the immunotolerant environment at the fetal-maternal interface (e.g., *CXCL16*, *HAVCR2/TIM-3*, *IL6*, *IL27RA*, *MADCAM1*, *TNFSF10*, *CSF1*, and *TRIM29*).

Taken together, gene expression profiling indicates that ZBED1 is implicated in various biological processes of trophoblasts. Most importantly, ZBED1 might downregulate genes supporting trophoblast differentiation and fusion but upregulate genes involved in the immunotolerant environment at the fetal-maternal interface.

## 4. Discussion

ZBED1 is a transcription factor with important roles in Drosophila differentiation, and this protein is also highly expressed in human trophoblast cells. Thus, the overall aim of this study was to gain a better understanding of the functions of ZBED1 in trophoblast differentiation and other processes related to placental biology. To this end, we used forskolin-induced differentiation of BeWo cells as a model as this recapitulates various aspects of trophoblast biology [13,14,15,25,39].

Our results indeed suggest that ZBED1 has an important role in trophoblast differentiation. First, many of the ZBED1 regulated genes are also regulated by forskolin, and *ZBED1* expression is enhanced upon forskolin treatment, both implicating ZBED1 in trophoblast differentiation. Second, loss of ZBED1 induced extensive apoptosis in BeWo cells in response to forskolin treatment, but not in untreated cells, indicating that loss of ZBED1 perturbs the differentiation process and compromises cell viability. In agreement, *ZBED1* knockdown enhanced the expression of genes that generally induce apoptosis (e.g., *GADD45G* [40], *RELT* [41], *DDIT3* [42], *EGR1* [43], *KL4* [44] and *TNFRSF1B* [45]), and decreased the expression of genes supporting trophoblast proliferation (e.g., *CAV1* [46], *EPO* [47], *PDGFRB* [48], and *PRR15* [49]).

The effect of *ZBED1* knockdown on gene expression was most pronounced 24 h after forskolin treatment, suggesting that ZBED1 might be important in the early stages of trophoblast differentiation. In addition to significant induction of genes favoring differentiation and invasion (e.g., *APELA* [50], *BGN* [51], *HSPB1* [52], *INHA* [53], *KLF4* [54], *SGK1* [55], and *TBX3* [56]), *ZBED1* knockdown also induced expression of syncytiotrophoblast markers (e.g., *ERVFRD-1* [17], *CYP19A1* [20], and *DYSF* [19]) and repressed cytotrophoblast makers (e.g., *ID-1* [57], *CAV1* [46,58], *ELF-5* [59], and *NR6A1* [60]) that also inhibit differentiation and are crucial for cytotrophoblast maintenance. Normally these genes are highly expressed in CTBs, but their expression diminishes during differentiation. Moreover, *ELF-5* might be important for regulating the switch between trophoblast stem cell proliferation and differentiation [59]. Together, these results indicate that ZBED1 might be important for maintaining the steady pool of proliferating cytotrophoblast cells, though additional experiments are needed to confirm this.

Trophoblasts actively recruit leukocytes and modulate their functional profile in order to maintain immune tolerance of the placenta by producing various chemokines, cytokines, and adhesion molecules [26]. Interestingly, ZBED1 was found to regulate genes implicated in immune-related processes. More specifically, *ZBED1* knockdown decreased the expression of CXCL16 and MAdCAM-1, both of which attract maternal macrophages [61,62,63], and *IL-6* and *CSF-1*, which has been shown to not only recruit leukocytes but also promote the differentiation of monocytes and T cells into an M2-like macrophage and T-regs, respectively [27,28,64]. ZBED1 was also found to support the expression of proteins that keep effector immune cells in check in the intervillous space (e.g., *TNFSF10/TRAIL* [30]) and regulate the expression of several genes important for leukocyte migration and immune responses in general (e.g., *TNFSF12* [65], *TSC22D3* [66], *TRIM29* [67], and *HAVCR2/TIM-3* [68]), further emphasizing an immunoregulatory role of ZBED1.

Various intracellular signaling cascades have been associated with stimulating syncytiotrophoblast differentiation, including several pathways regulated by elevated cAMP (e.g., cAMP/PKA/CREB, cAMP/Epac1/CaMK1, and cAMP/p38MPK/ERK1/2 [19,23,69]), and the Wnt/β-catenin pathway [23]. Interestingly, *ZBED1* knockdown enhanced the expression of *HSPB8*, an activator of the ERK/CREB signaling pathway [70], and decreased the expression of *EID-2*, an inhibitor of EP300 [71] that is a co-activator of CREB. ZBED1 also regulated the Wnt pathway-associated *CAV1*, which induces cell cycle progression, cell proliferation, and anti-apoptosis signals [72], and *WNT11* and *FZD4*, which are important for blastocyte implantation and early pregnancy [73].

Generally, trophoblasts and cancer cells have a lot in common as trophoblasts mimic several malignant features, including angiogenesis, invasion of tissue, and avoiding immune responses [74]. Although this study was conducted based on a model of trophoblast differentiation, the BeWo cell line is a choriocarcinoma cell line, which emphasizes that the results obtained in this study could, in part, translate to cancer biology. Additionally, a recent study has demonstrated that *ZBED1* is overexpressed in gastric cancer tissues [6]. The study shows that ectopic expression of *ZBED1* significantly promoted cell proliferation and colony formation abilities of the gastric cancer cell lines, which was linked to a poor prognosis. Moreover, other zinc finger BED-type transcription factors, such as ZBED3 and ZBED6, are implicated in cancer progression in lung cancer cells [75] and colorectal cancer cells [76], respectively. As ZBED1 might be involved in regulating immune responses, and especially in promoting immunosuppressive functions at the fetal-maternal interface and might be linked to proliferation and apoptosis, it would be interesting to further investigate the role of ZBED1 in cancer-related settings.

The BeWo choriocarcinoma-derived cell line is widely used and accepted as a model for studying the biological processes of trophoblast cells, as it, to a large extend, retains the molecular profiles and phenotypes of normal trophoblast cells. However, the carcinogenic nature of these cells may infer limitations to the interpretations of our results. Thus, our results should be supplemented by future studies addressing the role of ZBED1 in trophoblast cell biology, using non-carcinogenic models such as in vitro cultured human trophoblast stem cells [77].

## 5. Conclusions

In conclusion, our study demonstrates a role for ZBED1 in the regulation of genes implicated in major biological processes of placental biology, including a potentially important role in maintaining a pool of proliferating cytotrophoblast cells and maternal immune tolerance.

## Figures and Tables

**Figure 1 genes-13-00133-f001:**
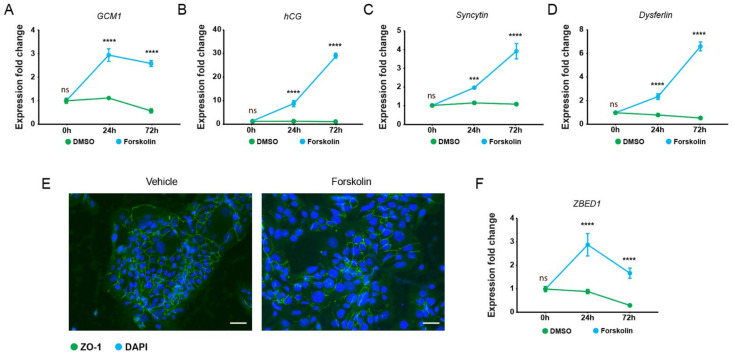
ZBED1 expression is induced during BeWo cell differentiation. (**A**–**D**) qPCR results of the relative gene expression of GCM1, hCGβ, syncytin-1, and dysferlin genes in BeWo cells (shLuc transfected; control for the *ZBED1* knockdown experiments) in response to DMSO (control) and forskolin. The data are presented as mean ± SEM. (**E**) Immunocytochemical staining of membrane protein ZO-1 (green) and DAPI (blue). Images were taken with 20× magnification. Scale bars = 50 µM. (**F**) qPCR analysis of *ZBED1* expression BeWo cells in response to DMSO (control) and forskolin. The data are presented as mean ± SEM. Two-way ANOVA with the following multiple comparison test of the mean values of the individual time points was used to calculate significance (*n* = 3). ns = non-significant. *** *p* < 0.0001; **** *p* < 0.00001.

**Figure 2 genes-13-00133-f002:**
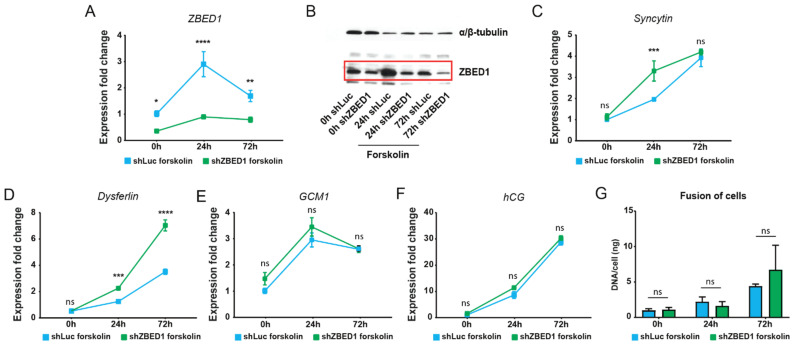
*ZBED1* knockdown regulates key factors of BeWo differentiation. (**A**,**B**) Validation of *ZBED1* knockdown in BeWo cells in response to forskolin by qPCR and Western blotting. (**C**–**F**) qPCR results of the relative gene expression of syncytin-1, dysferlin, GCM1, and hCGβ of BeWo cells with and without *ZBED1* knockdown (shZBED1 and shLuc, respectively) in response to forskolin. (**G**) Quantification of fusion by DNA/cell calculation. The data are presented as mean ± SEM. Two-way ANOVA with the following multiple comparison tests of the mean values of the individual time points was used to calculate significance (*n* = 3). ns = non-significant. **** *p* < 0.00001; *** *p* < 0.0001; ** *p* < 0.001; * *p* < 0.01.

**Figure 3 genes-13-00133-f003:**
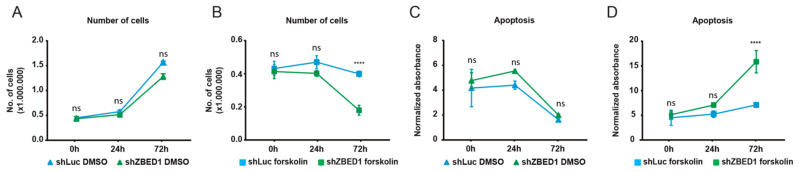
*ZBED1* knockdown induces apoptosis and reduces BeWo cell viability. (**A**,**B**) BeWo cells with and without *ZBED1* knockdown (shZBED1 and shLuc, respectively) were counted at 0, 24, and 72 h upon DMSO or forskolin treatment. (**C**,**D**) Induction of apoptosis in BeWo cells with and without *ZBED1* knockdown (shZBED1 and shLuc, respectively) were measured at 0 h, 24 h, and 72 h upon DMSO or forskolin treatment. The data are presented as mean ± SEM. Two-way ANOVA with the following multiple comparison test of the mean values of the individual time points was used to calculate significance (*n* = 3). ns = non-significant. **** *p* < 0.00001.

**Figure 4 genes-13-00133-f004:**
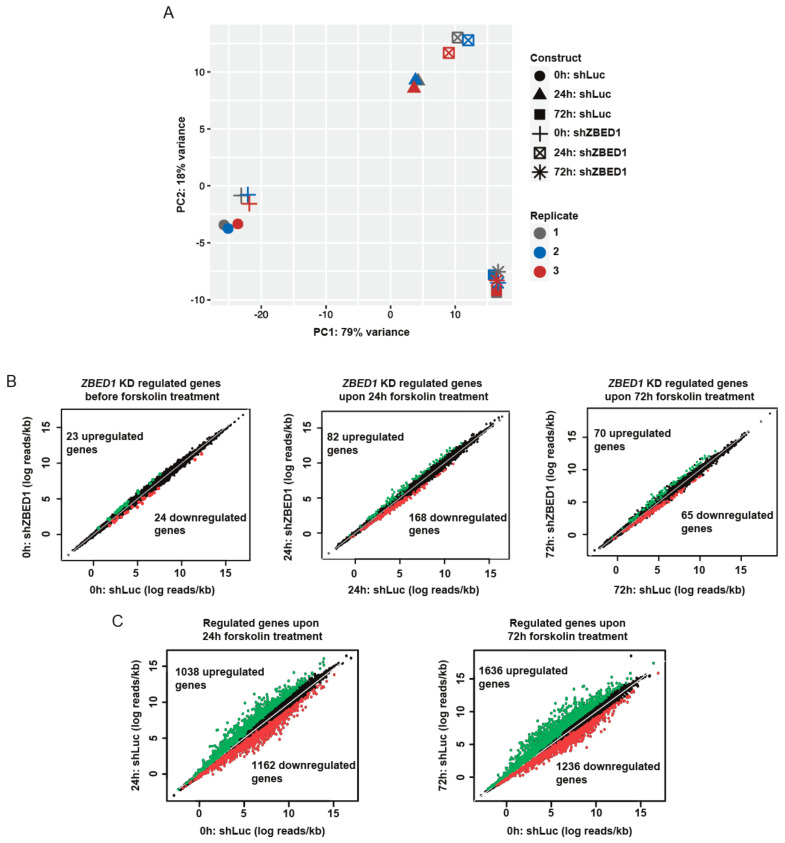
ZBED1 knockdown counteracts forskolin gene regulation. RNA-seq analysis of shLuc- and shZBED1-transduced BeWo cells in response to forskolin. (**A**) PCA-plot showing distinct transcriptional profiles of shZBED1- and shLuc-transduced cells in response to forskolin at 0 and 24 h, but not at 72 h. (**B**,**C**) Genes differentially expressed between shZBED1- and shLuc-transduced cells are shown. FDR-adjusted *p*-values were ≤0.05 and log2 fold change above and below 1 and −1, respectively.

**Figure 5 genes-13-00133-f005:**
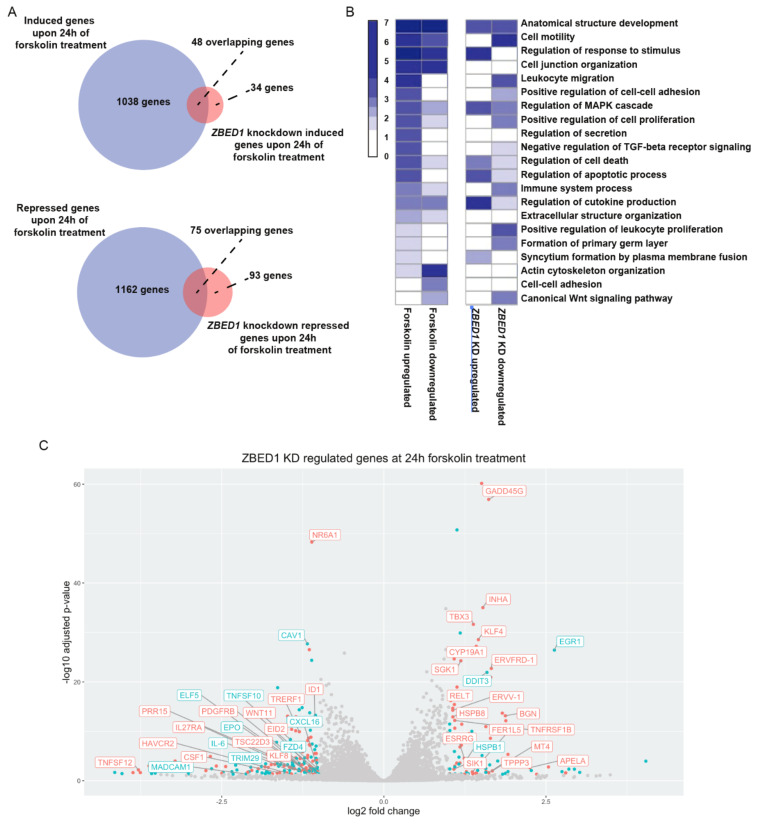
ZBED1 regulates crucial genes of biological processes related to BeWo differentiation. (**A**) Overlap of genes regulated by forskolin treatment (purple) and ZBED1 knockdown during forskolin treatment (pink) (RNA-seq data). (**B**) Selected biological processes of genes regulated by forskolin and ZBED1 knockdown. As the color-coding of the heatmap is based on adjusted *p*-values of groups of genes that differ significantly (forskolin vs. ZBED1 regulated genes), the color intensities cannot be compared between the forskolin and ZBED1 regulated genes, only within the groups. (**C**) Volcano plot of genes regulated by ZBED1 knockdown (green dots) that also overlap with genes regulated by forskolin (red) at 24 h.

## Data Availability

Further data will be made available upon request.

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
