# Peer review of "ZBED1 Regulates Genes Important for Multiple Biological Processes of the Placenta"

_genes, 2022, doi:10.3390/genes13010133_

Round 1
Reviewer 1 Report
In this manuscript the authors report on the contributions of ZBED1 to trophoblast cell biology. The experiments were performed using BeWo cells, a choriocarcinoma-derived cell line. Based on data generated from some functional experiments following shRNA-mediated ZBED1 knockdown and RNA-sequencing, they conclude that ZBED1 is important for trophoblast cell survival and maternal immune tolerance. However, there are issues with data presentation and the conclusions drawn from the data as stated. Individual comments and questions are outlined below.
- All work was completed in a choriocarcinoma-derived cell line, which, due to the nature by which the line was derived, can confound interpretation of results. Human trophoblast stem cell models are accessible and can readily and reproducibly differentiate into the villous trophoblast lineage.
- If ZBED1 expression is low prior to forskolin treatment (with no effects on proliferation and apoptosis and modest effects on the transcriptome in the absence of forskolin-Figs 3,4) but induced following syncytialization, can the authors clarify why the emphasis is put on a role in maintaining cytotrophoblast cells?
- The authors refer to hCG expression in Figs 1 and 2, but do not indicate which subunit(s). The qPCR primer is labeled at hCG-beta in the methods; however, it is unclear which transcript(s). The beta subunit of CG is encoded by 6 genes. Similarly, the label “Syncytin” is confusing as there are two syncytins. The methods indicate syncytin-1, but this needs to be more clearly indicated in the figure and in the text.
- BeWo cell fusion and induction of GCM1, Syncytin-1, etc. following forskolin treatment has previously been described and characterized.
- Were CP19A1 and Syncytin-2 also induced in this model with forskolin treatment?
- Figs 1-3: It is difficult to distinguish between the treatment groups based on the appearance of the data points on the graphs.
- Fig 1: Images need scale bars and labels in the figure to indicate what the colors represent. The two images appear as though they were taken at different magnification, although the legend indicates they are both from 20X. Scale bars would help clarify.
- Figs 1 and 2: Y axis labels are confusing. Do these values represent the relative expression or the fold change? These are two distinct values.
- Lines 233/234- if there was not a statistically significant difference, then don’t state that the DNA/cell ratio was increased.
- Do the relatively small changes in Dysferlin and Syncytin expression at 1-2 timepoints support the statement that ZBED1 plays an important role in regulating the expression of core trophoblast differentiation genes, especially in the absence of a difference in the cell fusion index?
- Does the ZBED1 knockdown data shown in Figure 2 apply to the proliferation and apoptosis data in Figure 3? If not, please show extent of ZBED1 knockdown in cells used for these studies.
- Fig 3: Since differences in several readouts weren’t observed until 72h, would it be worthwhile to look at later timepoints?
- Expression and functional validation of RNA-seq data would strengthen the manuscript.
Author Response
RE: Genes -1521211
Dear editors and reviewers,
Thank you very much for considering our manuscript “ZBED1 regulates genes important for multiple biological processes of the placenta” by Johansen et al. and for your invitation to submit a revised version. We were delighted to see that our work was well received by the reviewers, and we greatly appreciate the opportunity to respond to the relevant and constructive criticism. We believe that we have successfully addressed the concerns raised by the reviewers and we have included a point-by-point response below.
We feel that the revisions have significantly strengthened our study and hope that our manuscript is now suitable for publication in Genes.
Yours sincerely,
Morten Gjerstorff
Associate professor
Reviewer 1
- All work was completed in a choriocarcinoma-derived cell line, which, due to the nature by which the line was derived, can confound interpretation of results. Human trophoblast stem cell models are accessible and can readily and reproducibly differentiate into the villous trophoblast lineage.
Response: We agree with the reviewer that the choice of cell model will affect the interpretation of results, which is of course the case for all studies relying on models. The BeWo choriocarcinoma-derived cell line is widely used and accepted as a model for studying the biological processes of trophoblast cells, as it, to a large extend, retains the molecular profiles and phenotypes of normal trophoblast cells (Hannan et al., Biology of Reproduction, 2010). Furthermore, BeWo cells are well characterized and have been used in numerous studies, which permits a wide interpretation of our results. The reviewer mentions human trophoblast stem cells as a possible model for studying the cellular and biological functions of ZBED1. Indeed, it has recently been demonstrated that human trophoblast stem cells can be cultured and differentiated in vitro. However, to our knowledge such models remain poorly characterized and are not easily available. Thus, we believe that BeWo cells remains the best choice for our studies.
- If ZBED1 expression is low prior to forskolin treatment (with no effects on proliferation and apoptosis and modest effects on the transcriptome in the absence of forskolin-Figs 3,4) but induced following syncytialization, can the authors clarify why the emphasis is put on a role in maintaining cytotrophoblast cells?
Response: As stated by the reviewer, the ZBED1 expression is reduced upon forskolin-treatment, but ZBED1 expression is not “low” in BeWo cells prior to forskolin treatment, as demonstrated by Western blotting and quantitative PCR (Figure 2). In agreement, we have previously shown that ZBED1 is expressed in both cytotrophoblast and syncytiotrophoblasts (Hansen et al. Plos One, 2018). Our emphasis on the potential role for ZBED1 in maintaining cytotrophoblast cells originates from the results of our RNA sequencing analysis implicating ZBED1 in repression of multiple genes implicated in trophoblast differentiation.
- The authors refer to hCG expression in Figs 1 and 2, but do not indicate which subunit(s). The qPCR primer is labeled at hCG-beta in the methods; however, it is unclear which transcript(s). The beta subunit of CG is encoded by 6 genes. Similarly, the label “Syncytin” is confusing as there are two syncytins. The methods indicate syncytin-1, but this needs to be more clearly indicated in the figure and in the text.
Response: We agree with the reviewer and have adjusted the manuscript accordingly (Lines 160, 196, 197, 220, 226 and 235)
- BeWo cell fusion and induction of GCM1, Syncytin-1, etc. following forskolin treatment has previously been described and characterized.
Response: Indeed, the BeWo cell model is well characterized and widely used, which supports our choice of this cell model.
- Were CP19A1 and Syncytin-2 also induced in this model with forskolin treatment?
Response: Yes, both genes are upregulated by forskolin treatment, as seen in the volcano plot (CYP19A1 and ERVFRD-1, respectively; Figure 5). They are both regulated by ZBED1 (upregulated by knockdown) and regulated by forskolin (upregulated upon forskolin stimuli).
- Figs 1-3: It is difficult to distinguish between the treatment groups based on the appearance of the data points on the graphs.
Response: We agree, it is hard to distinguish the groups and we have changed the graphs accordingly (Figure 1-3).
- Fig 1: Images need scale bars and labels in the figure to indicate what the colors represent. The two images appear as though they were taken at different magnification, although the legend indicates they are both from 20X. Scale bars would help clarify.
Response: We have included scale bars (Figure 1E). Upon forskolin stimuli, the nuclei become bigger, which is why it could seem that the pictures are taken with different magnification.
- Figs 1 and 2: Y axis labels are confusing. Do these values represent the relative expression or the fold change? These are two distinct values.
Response: The y-axis displays the fold change of the relative to day 0. This has been corrected (Figure 1 and 2).
- Lines 233/234- if there was not a statistically significant difference, then don’t state that the DNA/cell ratio was increased.
Response: We agree with the reviewer and has corrected the manuscript accordingly (lines 236).
- Do the relatively small changes in Dysferlin and Syncytin expression at 1-2 timepoints support the statement that ZBED1 plays an important role in regulating the expression of core trophoblast differentiation genes, especially in the absence of a difference in the cell fusion index?
Response: We believe that the reproducible and statistically significant changes in Dysferlin and Syncytin expression upon ZBED1 knockdown are of biological significance. These data should be seen in the context of our RNA sequencing analysis showing that multiple genes involved in trophoblast differentiation are regulated by ZBED1.
- Does the ZBED1 knockdown data shown in Figure 2 apply to the proliferation and apoptosis data in Figure 3? If not, please show extent of ZBED1 knockdown in cells used for these studies.
Response: We have not shown ZBED1 expression data for all our experiments. However, the data shown in figure 2 are indeed representative for the experiments in figure 3 and other figures.
- Fig 3: Since differences in several readouts weren’t observed until 72h, would it be worthwhile to look at later timepoints?
Response: We agree with the reviewer that readouts at later timepoint would be interesting. However, the effect of ZBED1 knockdown on cell survival during forskolin treatment favors the outgrowth of cells without ZBED1 knockdown, which hinders the analysis at later time points.
- Expression and functional validation of RNA-seq data would strengthen the manuscript.
Response: We believe that the absolute quantification of transcripts performed by RNA sequencing does not require following validation of expression and further functional experiments are beyond the scope of this manuscript.
Reviewer 2
This study is considered to have important implication about the roles ZBED1plays in trophoblast differentiation. However, the authors used only choriocarcinoma cell line BeWo cells. Considering malignant features as choriocarcinoma, it is not necessarily possible to suggest that the biological functions and gene expression showed in this study could be obtained purely from trophoblast features. Validation using normal trophoblast cell line is desirable.
Response: We agree with the reviewer that the choice of cell model is associated with limitations to the interpretation of the results, which is the case with any cell model. However, the BeWo cell line model is well validated and used in numerous studies. Validating our data using “normal trophoblast cell lines” would indeed be preferable, but to our knowledge such cell lines is not available. Please see the response to reviewer 1, comment 1 for further details.
Reviewer 2 Report
With a view to gain a better understanding of the functions of ZBED1 in trophoblast differentiation, the authors investigated the biological functions of ZBED1 in trophoblast differentiation using forskolin-induced BeWo cell fusion model and ZBED1-knockdown BeWo cells. They showed increase in apoptosis upon knockdown of ZBED1 and implication of ZBDE1 in various biological processes of trophoblasts using functional enrichment analysis from RNA sequencing of BeWo cells with or without ZBED1 knockdown upon forskolin treatment.
Comments
This study is considered to have important implication about the roles ZBED1plays in trophoblast differentiation. However, the authors used only choriocarcinoma cell line BeWo cells. Considering malignant features as choriocarcinoma, it is not necessarily possible to suggest that the biological functions and gene expression showed in this study could be obtained purely from trophoblast features. Validation using normal trophoblast cell line is desirable.
Author Response

(The authors gave the same response as above.)

Round 2
Reviewer 1 Report
Human trophoblast stem cell models are superior and quite widely used/available since the initial publication (Okae et al.) in late 2017. As a result, it is becoming increasingly difficult to publish results obtained solely from immortalized or choriocarcinoma-derived trophoblast cell lines, such as the BeWo cells used in these studies. Reviewer 2 shared the same concern. To try to address this signifiant shortcoming, I recommended that the authors provide some commentary in their discussion acknowledging the limitations of the model they utilized for these studies.
Reviewer 2 Report
As authors commented, BeWo cell have been well investigated and many works were reported using this cell line indeed. I agree the idea using BeWo cells to investigate the role of ZBED1 in trophoblast cell as a first step. However, previously one of the main reason why this cell line chosen was the difficulty of getting stable primary trophoblast cells and nowadays it become possible to use or obtain stable normal trophoblast from its stem cell. So, I agree accepting this article in anticipation of authors’ next work using normal trophoblasts for getting closer to the truth.